

**Molecular characteristics and diurnal variations of organic aerosols**
**at a rural site in the North China Plain with implications for the**
**influence of regional biomass burning**
Jianjun Li[1,4], Gehui Wang[1,2,3] *, Qi Zhang[4]*, Jin Li[1], Can Wu[2], Wenqing Jiang[4], Tong
Zhu[5], and Limin Zeng[5]
[1] Key Lab of Aerosol Chemistry & Physics, SKLLQG, Institute of Earth Environment,
Chinese Academy of Sciences, Xi'an 710061, China
[2]Key Laboratory of Geographic Information Science of the Ministry of Education,
School of Geographic Sciences, East China Normal University, Shanghai 200241,
China
[3]Institute of Eco-Chongming, 3663 N. Zhongshan Rd., Shanghai 200062, China
[4]Department of Environmental Toxicology, University of California, Davis, CA 95616,
USA
[5]BIC-ESAT and SKL-ESPC, College of Environmental Sciences and Engineering,
Peking University, Beijing, China
*Corresponding authors:
Prof. Gehui Wang, E-mail: ghwang@geo.ecnu.edu.cn;
Prof. Qi Zhang, E-mail: dkwzhang@ucdavis.edu



**Abstract**
Field burning of crop residue in early summer releases into the atmosphere a
large amount of pollutants with significant impacts on the air quality and aerosol
properties in the North China Plain (NCP). In order to investigate the influence of this
regional anthropogenic activity on organic molecular characteristics of aerosol, we
collected $PM_{2.5}$ filter samples every 3 hours at a rural site of NCP during June 10th to
25th, 2013, and analyzed them for more than 100 organic tracer compounds, including
both primary ($n$-alkanes, fatty acids/alcohols, sugar compounds, polycyclic aromatic
hydrocarbons, hopanes, and phthalate esters) and secondary (phthalic acids, isoprene-,
α-/β-pinene, β-caryophyllene, and toluene-derived products) organic aerosol tracers,
as well as for organic carbon (OC), elemental carbon (EC), and water-soluble organic
carbon (WSOC). Total concentrations of the measured organics ranged from 177 to
6248 ng m$^{-3}$ (mean 1806±1308 ng m$^{-3}$) during the study period, most of which were
contributed by sugar compounds, followed by fatty acids and fatty alcohols.
Levoglucosan (240±288 ng m$^{-3}$) was the most abundant single compound and
strongly correlated with OC and WSOC, suggesting that biomass burning (BB) is an
important source of summertime organic aerosols in this rural region. Based on
analysis of fire spots and backward trajectories of air masses, two representative
periods were classified, which are (1) Period 1 (P1), Jun 13th 21:00-16th 15:00, when
air masses were uniformly from the southeast part of NCP, where intensified
open-field burning of biomass fuels occurred and (2) Period 2 (P2), Jun 22nd
12:00-24th 06:00, which were representative of local emission. Nearly all the



measured PM components showed much higher concentrations in P1 than in P2.
Although *n*-alkanes, fatty acids, and fatty alcohols presented similar temporal/diurnal
variations as those of levoglucosan throughout the entire period, their molecular
distributions were more dominated by high molecular weight (HMW) compounds in
P1, demonstrating an enhanced contribution from BB emissions. In contrast,
intensified BB emission in P1 seems to have limited influences on the concentrations
of polycyclic aromatic hydrocarbons (PAHs), hopanes and phthalate esters. Both
3-hydroxyglutaric acid and β-caryophyllinic acid showed strong linearly correlations
with levoglucosan ($R^2$=0.72 and 0.80, respectively), indicating that biomass burning
is also an important source for terpene-derived SOA formation. A tracer-based method
was used to access the distribution of biomass-burning OC, fungal-spore OC and
secondary organic carbon (SOC) derived from isoprene, α-/β-pinene, β-caryophyllene,
and toluene in the different periods. The results showed that the contribution of
biomass-burning OC to total OC in P1 (27.6%) was 1.7 times of that in P2 (17.1%).
However, the contribution of SOC from oxidation of the four kinds of VOCs
increased slightly from 16.3% in P1 to 21.1% in P2.
***Key words***: Organic aerosols; Molecular composition; North China Plain; Biomass
Burning



# 1. Introduction

Organic aerosols (OA, i.e., the organic fraction of particles) constitute a substantial fraction (~10-90%) (Jimenez et al., 2009;Zhang et al., 2007;Hallquist et al., 2009) of atmospheric particles, and have significant effects on global and regional climate (Venkataraman et al., 2005;Kanakidou et al., 2005), air quality (Aggarwal et al., 2013;Wang et al., 2006b), human health (Lelieveld et al., 2015), and ecosystems (Tie et al., 2016). Organic aerosols in the atmosphere can be emitted directly from various sources, such as combustion of fossil fuels, biomass burning, plant emission, and so on, which is defined as primary organic aerosols (POA). On the other hand, atmospheric secondary OA (SOA) are produced from photochemical oxidation products of volatile organic compounds (VOCs) via gas-particle conversion processes such as nucleation, condensation and heterogeneous chemical reactions (Hallquist et al., 2009). These organic species could modify physicochemical characteristics of atmospheric aerosols such as hygroscopicity, albedo, and oxidation state (Dinar et al., 2008;Chan et al., 2005;Fu et al., 2010). Thus, a thorough understanding of molecular composition and sources of organic aerosols is necessary in order to address aerosol related environmental issues and to improve the accuracy of modelling studies.

Tremendous amounts of air pollutants including both particulate matters (PM) and their gaseous precursors (e.g., $SO_2$, $NO_x$, $NH_3$, and VOCs) are emitted into the atmosphere from power plants, industries and vehicles due to rapid economy development in China, leading to serious air conditions in the recent decades (Zhang et al., 2009;Guo et al., 2014;Wang et al., 2016;Huang et al., 2014;Li et al., 2017). The





North China Plain (NCP) has been recognized as one of the most polluted regions in
the world, with very high concentrations of $PM_{2.5}$ on the ground surface (van
Donkelaar et al., 2010). The NCP is also considered as one of the most significant
aerosol sources, which has a significant impact on the East China Sea and western
North Pacific (Andreae and Rosenfeld, 2008). Thus, extensive efforts have been made
in recent years to characterize the sources, properties, and processes of PM in the NCP.
Most of these results concluded that the severe air pollution in the region is related to
the source strength and frequently happens under stagnant weather conditions.
Recently, it has been shown that the exponential growth of secondary aerosols could
lead to an extreme haze event under certain meteorological conditions (Wang et al.,
2016;Sun et al., 2014;Quan et al., 2013).
In the rural area of NCP, biomass burning for domestic cooking and heating and
agricultural waste disposal is an important source of atmospheric PM (Wang et al.,
2009b;Li et al., 2010;Zhang et al., 2016). Particularly, the open field burning is still a
common way for disposal of agricultural residues (mainly wheat straws) in early
summer (Li et al., 2007). This traditional activity could release huge amount of
pollutants into the atmosphere and significantly affect air quality and aerosol
properties in the region. Zhu et al. (2016) examined the amounts of VOCs in the air at
a rural site of Yucheng (Shandong Province, East China), and found that their
concentrations during the wheat straw burning period are approximately twice of
those in normal periods. Model results also revealed a significant influence of open
crop residual burning on ozone, CO, black carbon (BC) and organic carbon (OC)





concentrations in NCP. Moreover, both off-line (Fu et al., 2012) and on-line (Sun et
al., 2016) observations indicated that the intensified emission from wheat straw
burning in the region could change the molecular distribution of organic aerosols of
the downwind urban or mountain areas.
During June $10^{th}$ to $25^{th}$ of 2013, we conducted a continuous sampling campaign
at a rural site in the northern part of NCP. $PM_{2.5}$ filter samples were continuously
collected with a 3-hour time resolution and determined for more than 100 organic
compounds including aliphatic lipids, sugar compounds, hopanes, polycyclic aromatic
hydrocarbons (PAHs), phthalate esters, and secondary oxidation products. The first
objective of this study is to get an overall understanding of temporal/diurnal variation
and molecular distribution of summertime OA in the rural region. The second
objective is to compare the results in two representative periods to investigate the
influence of regional field burning of wheat straw on the molecular characteristics of
organic aerosols.
**2. Experimental section**
**2.1 Sample collection**
The measurement was performed at the Integrated Ecological-Meteorological
Observation and Experiment Station of Chinese Academy of Meteorological Sciences
(39°08' N, 115°40' E, 15.2 m asl), which is located in a rural area of Gucheng, Hebei
Province. Detailed information of the station and sampling campaign was described in
Li et al. (2018). Briefly, $PM_{2.5}$ samples were collected on the rooftop (about 10 m
above the ground) of a three-story building on the campus of the Gucheng station.



The sampling was conducted from June 10th to 25th, 2013 by using a high volume
(1.13 $m^3$ $min^{-1}$) sampler (Anderson) with a three hour of duration in each. All samples
were collected onto pre-baked (450 °C, 6-8 hr) quartz fiber filters. Field blank samples
were also collected by mounting blank filters onto the sampler for about 15 min
without pumping any air. After sampling, the sample filter was individually sealed in
aluminum foil bags and stored in a freezer (−20 °C) prior to analysis.

**142    2.2 Organic compounds determination**

A size of 12.5-25 $cm^2$ of the filter sample was cut and extracted with a mixture of
dichloromethane and methanol (2:1, v/v) under ultrasonication. The extracts were
concentrated using a rotary evaporator under vacuum conditions and then blow down
to dryness using pure nitrogen. After reaction with N,O-bis-(trimethylsilyl)
trifluoroacetamide (BSTFA) at 70 ℃ for 3 hrs., the derivatives were determined
using gas chromatography/electron ionization mass spectrometry (GC/EI-MS) (Li et
al., 2013b).
Gas chromatograph/mass spectrometry (GC/MS) analysis of the derivatized
fraction was performed using an Agilent 7890A GC coupled with an Agilent 5975C
MSD. The GC separation was carried out on a DB-5MS fused silica capillary column
with the GC oven temperature programmed from 50℃ (2 min) to 120℃ at 15℃
$min^{-1}$ and then to 300℃ at 5℃ $min^{-1}$ with a final isothermal hold at 300℃ for 16
min. The sample was injected in a splitless mode at an injector temperature of 280℃,
and scanned from 50 to 650 Daltons using electron impact (EI) mode at 70 eV.
GC/MS response factors of all the target compounds were determined using





authentic standards except several isoprene-derived SOA tracers. Response factors of
isoprene-derived SOA tracers were substituted by those of related surrogated
standards, which were described in Li et al. (2018). No significant contamination (<5%
of those in the samples) was found in the blanks. Recoveries of all the target
compounds ranged from 80% to 120%. Data presented were corrected for the field
blanks but not corrected for the recoveries.
**2.3 OC, EC, and WSOC analysis**

OC (organic carbon) and EC (elemental carbon) were analyzed using DRI Model

2001 Carbon Analyzer following the Interagency Monitoring of Protected Visual
Environments (IMPROVE) thermal/optical reflectance (TOR) protocol. A size of
0.526 cm$^2$ sample filter was placed in a quartz boat inside the analyzer and stepwise
heated to temperatures of 140 ℃ (OC1), 280 ℃ (OC2), 480 ℃ (OC3), and 580 ℃
(OC4) in a non-oxidizing helium (He) atmosphere, and 580 ℃ (EC1), 740 ℃
(EC2), and 840 ℃ (EC3) in an oxidizing atmosphere of 2% oxygen in helium.
Pyrolyzed carbon (PC) is determined by reflectance and transmittance of 633 nm light.
One sample was randomly selected from every 10 samples and re-analyzed.
Differences determined from the replicate analyses were <5% for TC, and <10% for
OC and EC.

Another aliquot of filter sample was extracted with organic-free Milli-Q water

under ultrasonication (15 min each, repeated 3 times) and filtered through a PTFE
filter to remove any particles and filter debris. Then the water-extract was analyzed
for water-soluble organic carbon (WSOC) using a TOC analyzer (TOC-L CPH,



Shimadzu, Japan). The difference between OC and WSOC was considered as
water-insoluble OC (WIOC). All carbonaceous components data reported here were
corrected by the field blanks.

**3. Results and discussion**

**3.1 Fire spots and air masses**

At present, open-field burning is still a common activity for disposal of crop
residue in the rural area of the North China Plain, especially during wheat harvest
period from the end of May to the middle of June (Fu et al., 2012). These extensive
emissions from regional biomass burning in the provinces of Anhui, Jiangsu,
Shandong, Henan and Hebei in the NCP can cause severe air pollution on a local and
regional scale. In our previous study, the fire spots in the North China during the
sampling period were provided based on the NASA satellite observation
(https://firms.modaps.eosdis.nasa.gov/firemap/). Combining with information on air
mass back-trajectories (http://ready.arl.noaa.gov/HYSPLIT.php), the sampling period
was divided into two sections: (1) June 10-18, when air masses were mainly
transported via long distances from the southeast part of NCP where intensive
emissions from the wheat straw burning occurred; (2) June 19-25, when air masses
were mostly influenced by local emissions and regional emission from biomass
burning decreased dramatically (Li et al., 2018). In this study, we further select two
representative periods to access the contribution of regional biomass burning. Period 1
(P1) designates $13^{th}$ Jun 21:00 pm to $16^{th}$ Jun 15:00 pm, during which air masses were
influenced by intensive biomass burning and transported uniformly from the southeast





part of NCP (Figure 1 a and b, and Figure S1). Period 2 (P2) designates 22$^{nd}$ Jun
12:00 pm to 24$^{th}$ Jun 06:00 am, during which fire spots in the regions were relatively
scarce and air masses came predominantly from the surrounding areas of the sampling
site (Figure 1 c and d). In addition, there were several intermittent rainfalls during
June 20-22, which are favorable for wet deposition of atmospheric pollutants. Thus,
aerosols collected in P2 are well representative of local fresh emission. It is
worthwhile to note that the two samples collected during 21$^{st}$ June 18:00-24:00 pm
were excluded from P2, because they were highly affected by near-site biomass
burning emission (detailed information is provided in Section 3.3).
**3.2 Concentrations of PM$_{2.5}$, OC, EC, WSOC and WIOC**

Concentrations of PM$_{2.5}$ and carbonaceous components are presented in Table 1.

PM$_{2.5}$ concentrations range from 21 to 395 µg m$^{-3}$ with a mean value at $159\pm89$ µg
m$^{-3}$ during the whole sampling period. As shown in Figure 2, PM$_{2.5}$ concentrations in
P1 (average $\pm$ 1$\sigma$ = $231\pm89$ µg m$^{-3}$) increase continuously from around 150 µg
m$^{-3}$ to higher than 300 µg m$^{-3}$, indicating the occurrence of a severe air pollution
episode. In contrast, PM$_{2.5}$ concentration during P2 is as low as $43\pm14$ µg m$^{-3}$.
Similarly, the average concentration of OC is $29.4\pm7.8$ µg m$^{-3}$ in P1, which is more
than 5 time higher than that in P2 ($5.5\pm1.7$ µg m$^{-3}$) . EC concentrations also decrease
dramatically from P1 ($12.1\pm4.0$ µg m$^{-3}$) to P2 ($1.5\pm1.5$ µg m$^{-3}$). The average
OC/EC ratio is $3.0\pm0.9$ for the whole sampling period, but the ratio was higher in P2
($3.8\pm1.0$) than in P1 ($2.5\pm0.4$), mainly due to the high SOA formation activities in
the rural areas of NCP in summer.



224   As shown in Figure 2 and 3, the concentrations of WSOC show a consistent

225 temporal variation as those of OC ($R^2$=0.82), highlighting the fact that WSOC is an

226 important fraction of OC in this region. In addition, the average ratio of WSOC/OC is

227 higher during P1 (0.62±0.16) than during P2 (0.48±0.12), mainly due to enhanced

228 emissions of water-soluble organic compounds (such as sugars, fatty alcohols/acids)

229 from biomass burning during P1. Due to the favorable meteorological conditions,

230 concentrations of water-insoluble OC in P2 (3.0±1.3 µg m$^{-3}$) are also much lower

231 than those in P1 (10.3±4.4 µg m$^{-3}$).

232   The diurnal variation profiles of EC/OC and WSOC/OC are shown in Figure 4.

233 EC/OC is generally lower in daytime and the lowest value occurred during

234 12:00-15:00 pm, mainly due to enhanced daytime formation of SOC. Previous studies

235 have shown that secondary organic aerosols are mainly composed of water-soluble

236 compounds, e.g., polyacids/polyalcohols and phenols (Kondo et al., 2007;Wang et al.,

237 2009a). However, these compounds can come from primary emissions as well,

238 especially from biomass burning (Shen et al., 2017;Fu et al., 2012). In this study, the

239 WSOC/OC presents lower value during daytime, especially in the afternoon when

240 photo-chemical oxidation is favorable. In addition, the diurnal variation pattern of

241 WSOC/OC is similar to that of levoglucosan/OC. Given that levoglucosan is a

242 well-known marker of biomass burning emissions (Simoneit et al., 1999;Simoneit et

243 al., 2004a) (detailed discussions are given in Section 3.3), these results indicate that

244 particulate WSOC in the region is mostly contributed by direct emissions from

245 biomass burning in the summer.





### 3.3 Organic molecular composition

More than 100 organic species were detected in the aerosol samples, and their

concentrations are shown in Table 2 and S1. In this study, these organic compositions

are grouped into 10 compound classes based on functional groups and sources. Total

concentrations of the measured organics range from 177 to 6248 ng m$^{-3}$ (average =

1806$\pm$1308 ng m$^{-3}$) during the whole sampling period with the predominance of

sugar compounds, followed by fatty acids and fatty alcohols. The temporal variation

profiles of the determined organic groups are shown in Figure 5. Nearly all the

measured organic species, especially $n$-alkanes, fatty acids, fatty alcohols, sugar

compounds, and PAHs, show much higher concentrations in P1 than in P2 (Figure S2),

indicating an important influence of regional biomass burning on airborne organic

aerosols in NCP.

### 3.3.1 Biomass-burning tracers

As described in Section 3.1, intensified emissions of open biomass burning were

observed in the southern part of NCP during June 13-16 (P1), which is an important

reason for the severe regional air pollution during this period. Levoglucosan, which is

produced in large quantities during pyrolysis of cellulose, is a key tracer for biomass

burning emissions (Simoneit, 2002;Simoneit et al., 1999). As shown in Table 2,

levoglucosan is the most abundant single compound in the whole sampling period,

ranged from 5.6 to 1447 ng m$^{-3}$ with a mean concentration of 240$\pm$288 ng m$^{-3}$.

Levoglucosan shows good positive correlations with both OC ($R^2$=0.61) and WSOC

($R^2$=0.65) (Figure 3), confirming that biomass burning is an important source of both



aerosol OC and WSOC in the rural region of NCP during the sampling period. As
clearly shown in Figure 6, the concentrations of levoglucosan present a continual
increasing trend during P1 with a mean value of $404 \pm 344$ ng m$^{-3}$. However, the tracer
presents very low concentrations (11-123 ng m$^{-3}$) for the most of time during Jun
20-22. Interestingly, the concentration of levoglucosan suddenly increased by more
than 10 times at 21$^{st}$ Jun 18:00 pm to approximately 1200 ng m$^{-3}$ in less than 3 hours
and decreased to its beginning concentration (less than 100 ng m$^{-3}$) within 6 hours (2
samples) afterwards. The concentrations of OC, WSOC and EC also showed obvious
peaks during this event. However, based on analyses of back-trajectories (Figure 1c)
and wind conditions (Figure S1), we didn't find significant change of air masses
origins. Also, not all organic markers showed similar variation as levoglucosan,
especially the concentrations of PAHs, hopanes, and phthalate esters changed little
during this event. Thus, it is plausible to conclude that this variation was caused by
emissions from biomass burning activities nearby the sampling site. For this reason,
the data of the 2 samples were excluded from P2.

The two isomers of levoglucosan, galactosan and mannosan, are also produced

by the pyrolysis of cellulose/hemicelluloses (Simoneit, 2002), and thus also
considered as important markers of biomass burning. Similar to levoglucosan, the
concentrations of these two anhydrosugars in P1 are 5-6 times higher than those in P2.
The isomeric ratios of levoglucosan to other anhydrosugars are considered as good
indicators of straw burning. For example, mannosan and galactosan were detected at
low levels in the smoke of lignites (Fabbri et al., 2009;Fabbri et al., 2008). As shown


in Table 2, the ratios of levoglucan/mannosan (L/M) and
levoglucan/(galactosan+mannosan) (L/G+M) both showed higher values in P1 than in
P2, agreeing well with the results reported by Fu et al. (2012) at Mt. Tai. These results
confirmed the great contribution of open burning of wheat straw to the organic
aerosols in the sampling region during P1.
**3.3.2 Aliphatic lipid compositions**

The average concentrations of all the *n*-alkanes ($C_{18}$–$C_{36}$) measured in this study

is $207\pm149$ ng m$^{-3}$ with the most abundant individual compound being nonacosane
($C_{29}H_{60}$), i.e., the carbon number maximum ($C_{max}$) is $C_{29}$ (Table S1). *n*-Alkanes
derived from terrestrial plants are dominated by high molecular weight species (HMW,
carbon number >25) with an odd number preference. In contrast, fossil fuel derived
*n*-alkanes do not have odd/even number preference (Rogge et al., 1993b;Simoneit et
al., 2004b). In general, *n*-alkanes with a carbon preference index (CPI, odd/even)
more than 5 are considered as plant wax, while those with a CPI nearly unity are
mostly derived from fossil fuel combustion (Rogge et al., 1993b, a). In this study, the
mean value of CPI is $2.47\pm1.12$, indicating that both fossil fuel and plant wax
contributed to *n*-alkanes in the rural areas of NCP in summer. However, *n*-alkanes
showed a stronger odd/even carbon number predominance in P1 (CPI=2.85) than in
P2 (CPI=1.64). In addition, all the low molecular weight *n*-alkanes (LMW, carbon
number <25) presented a higher contribution to total *n*-alkanes in P2 than in P1
(Figure 7 a and d). These results demonstrate that plant waxes from biomass burning
emissions made a bigger contribution to organic aerosols in the sampling region



during P1.

A homologous series of 19 saturated fatty acids ($C_{12:0}$–$C_{32:0}$) and 3 unsaturated

fatty acids ($C_{16:1}$, $C_{18:1}$, and $C_{18:2}$) were detected in the samples (Table S1), and their
total concentrations were $514\pm384$ ng m$^{-3}$ during the whole period. A strong even
carbon number predominance was observed with $C_{max}$ at $C_{28:0}$ and $C_{16:0}$ (Table S1).
Higher molecular weight (HMW) fatty acids ($\geqslant C_{20}$) are derived from terrestrial plant
waxes, while LMW fatty acids ($\leqslant C_{19}$) have multiple sources such as vascular plants,
microbes and marine phytoplankton as well as kitchen emissions (Rogge et al.,
1993a;Kawamura et al., 2003). The total concentrations of fatty acids presented
similar temporal variation to levoglucosan and well linearly correlated with it
($R^2$=0.72) (Figure 8a), indicating that fatty acids are mostly affected by biomass
burning emission during the whole sampling period. Still, there are some evidences
that regional emission from wheat straw burning significantly affected the distribution
of fatty acids in the aerosols of Gucheng during P1. Firstly, the total concentrations of
fatty acids in P1 ($900\pm358$ ng m$^{-3}$) are more than 6 times higher than those in P2
($145\pm48$ ng m$^{-3}$). Secondly, the concentrations and relative contributions of HMW
fatty acids ($C_{20:0}$–$C_{32:0}$) are much higher in P1 than in P2, similar to the results of
*n*-alkanes. In addition, the mean value of CPI of HMW fatty acids in P1 ($4.21\pm1.14$)
is also higher than that in P2 ($3.50\pm1.64$).

Fatty alcohols in the range of $C_{22}-C_{30}$ were detected for the PM$_{2.5}$ samples with a

mean concentration of $193\pm187$ ng m$^{-3}$ (Table 2 and S1) during the whole sampling
period. Their distributions are characterized by even carbon number predominance




with a maximum at $C_{28}$ (Figure 7c and f). HMW fatty alcohols ($\geqslant C_{20}$) abundantly
present in higher plants and loess deposits (Wang and Kawamura, 2005), thus the total
concentration of fatty alcohols strongly correlated with levoglucosan ($R^2$=0.73)
(Figure 8b). Similarly, nearly 10 times higher concentration of fatty alcohols was
observed in P1 (322$\pm$151 ng m$^{-3}$) compared with those in P2 (34$\pm$23 ng m$^{-3}$).
**3.3.3 Primary saccharides**
In addition to the three anhydrosugars, 4 primary sugars (fructose, glucose,
sucrose and trehalose) and 3 sugar alcohols (arabitol, mannitol and inositol) were
detected in the samples. Primary saccharides have been used as biomarkers for
primary biota emissions (Wang et al., 2011). Their mean concentrations ranged from
3.6 to 49 ng m$^{-3}$ during the whole sampling period. In this study, concentrations of
fructose, sucrose and trehalose in P1 were 7-10 times higher than those in P2 (Table
S1). They well correlated with levoglucosan ($R^2$=0.47-0.62, Figure S3) during P1, in
contrast to P2, during which no relationships were found between them. These results
indicated that these primary sugars were also affected by open-field emissions of
biomass burning during P1. Sugar alcohols, mainly arabitol and mannitol, are
abundant in airborne fungal spores (Graham et al., 2002). Some studies suggested that
biomass burning activities can enhance the emission of sugar alcohols at a certain
level (Engling et al., 2009;Fu et al., 2012). However, no significant relationship
($R^2$<0.10) can be found between these sugar alcohols and levoglucosan even in P1,
indicating the negligible contribution of biomass burning to the tracers in this study.
**3.3.4 PAHs, Hopanes and Phthalates**



As shown in Figure 5, the temporal variation of PAHs, hopanes, and phthalate
esters were clearly different to those of the molecular tracers for biomass burning,
especially in P1. In contrast to the continuous increase of sugars, fatty acids, fatty
alcohols, and n-alkanes during P1, the concentration of PAHs, hopanes, and phthalate
esters showed obvious day-night variations, indicating that biomass burning activities
contributed little on these species. Phthalates are widely used as plasticizers in
synthetic polymers or softeners in polyvinylchlorides (PVC) (Simoneit et al., 2004b)
and can be directly emitted from the matrix into the air as they are not chemically
bonded with the matrix. Six phthalate esters were detected in the sampling aerosols,
i.e., dimethyl (DMP), diethyl (DEP), diisobutyl (DiBP), butyl isobutyl (BiBP),
di-n-butyl (DnBP), and bis(2-ethylhexyl) (BEHP) phthalates (Table S1). The
concentrations of total detected phthalate esters in P1 ($112\pm33$ ng m$^{-3}$) are around 2
times only higher than those in P2 ($51\pm18$ ng m$^{-3}$). Hopanes are abundant in coal and
crude oils and enriched in lubricant oil fraction (Oros and Simoneit, 2000;Kawamura
et al., 1995).They can be emitted to the atmosphere from coal burning and/or internal
combustion of fuel in engines. Only two dominant hopanes,
17α(H),21β(H)-30-norhopane($C_{29\alpha\beta}$) and 17α(H),21β(H)-hopane($C_{30\alpha\beta}$), were detected
in all of the samples in this study. Their average concentration in P1 ($4.40\pm2.48$ ng
m$^{-3}$) is ~ 2.5 times of that in P2 ($1.81\pm0.31$ ng m$^{-3}$). Considering the much higher
concentrations of levoglucosan in P1 (on average ~ 8 times higher than P2), these
results again confirmed a limited influence of biomass burning on concentrations of
phthalate esters and hopanes in the aerosols in the rural region. Thus, there were no




significant concentration changes of the two species at $21^{st}$ Jun 18:00-24:00 pm, when
the air masses were highly affected by nearby biomass burning activities.

PAHs are the products of incomplete combustion of carbon-containing materials

and are of high toxicity and carcinogenicity (Halek et al., 2008;Sultan et al., 2001).
Previous studies indicated that PAHs are mainly emitted from coal burning and
vehicle exhaust in most areas of China (Wang et al., 2006a). However, it has been
reported that combustion of biomass materials can also contribute to the PAHs in the
atmosphere (Simoneit, 2002;Ge et al., 2012;Young et al., 2016). In this study, PAHs
correlated weakly with levoglucosan during the whole sampling period ($R^2$=0.27). Yet
the concentrations of total PAHs in P1 (18.6±11 ng m$^{-3}$) are nearly 8 times higher
than those in P2 (2.3±1.0 ng m$^{-3}$). These results mean that although the emission of
biomass burning is not the most important source for PAHs during the entire period,
the intensified regional burning of wheat straw in P1 can also enhance the PAHs
concentration in the atmosphere of Gucheng.

As shown in Figure 9, all the primary aerosol markers mentioned above showed

lower concentrations in daytime with lowest concentrations at afternoon (12:00-18:00
pm), consisting with the favorable diffusion conditions caused by high temperature
and planetary boundary layer (PBL). However, the day-night variation of PAHs,
hopanes, and phthalate esters are more obvious than other species, again confirming
the lower contribution of biomass burning to these organic compositions.
**3.3.5 Secondary organic aerosols (SOA) tracers**

Eight compounds were identified as isoprene oxidation products in the PM$_{2.5}$





samples, including two methyltetrahydrofuran diols, three $C_5$-alkene triols, two
2-methyltetrols, and 2-methylglyceric acid (Table S1). Detailed information about
formation and contribution of these compositions were discussed in our previous
paper (Li et al., 2018). The concentrations of total detected isoprene-derived products
are $112\pm86$ ng m$^{-3}$, with much higher concentration in P1 ($209\pm105$ ng m$^{-3}$) than in
P2 ($57\pm29$ ng m$^{-3}$).

*cis*-Pinonic acid (PNA), pinic acid (PA), 3-hydroxyglutaric acid (HGA) and

3-methyl-1,2,3-butanetricarboxylic acid (MBTCA) were detected as tracers for
α-/β-pinene oxidation in this study, and their concentration are shown in Table S1. The
concentration of total detected α-/β-pinene oxidation tracers are $66\pm31$ ng m$^{-3}$, with
MBTCA ($31\pm14$ ng m$^{-3}$) being the major compound during the whole sampling
period. PNA and PA are considered as first-generation products of α-/β-pinene
oxidation. They can be produced by further oxidation of carbonyl-substituted Criegee
intermediates formed by α-pinene ozonolysis (Jenkin et al., 2000;Ma et al., 2008), or
by OH oxidation of α-pinene under NO$_x$ free conditions (Eddingsaas et al.,
2012;Xuan et al., 2015). The formation of 3-HGA is supposed to be based on a ring
opening mechanism and may be related to a heterogeneous reaction of these
monoterpenes with irradiation in the presence of NOx (Jaoui et al., 2005;Claeys et al.,
2007). As shown in Figure 6b-d, PNA, PA and HGA present similar temporal
variations and correlated well with each other ($R^2$=0.48-0.76, Figure S4). The
formation of MBTCA is explained by further photodegradation of *cis*-pinonic acid
and pinic acid with OH radical (Müller et al., 2012;Szmigielski et al., 2007). As a



later-generation oxidation products, MBTCA showed an obviously different temporal
variation profile than those of PNA and PA, and had no significant increase during P1.
In addition, the concertation of PNA, PA and HGA in P1 are 2-8 times higher than
those in P2. However, the concentration of MBTCA in the two periods are
comparable. These results are consistent with the longer time scales of formation
pathway, lower volatility and longer lifetime of MBTCA in the atmosphere compared
to the first-generation products of α-/β-pinene oxidation. β-Caryophyllinic acid,
formed either by ozonolysis or photo-oxidation of β-caryophyllene (a sesquiterpene)
(Jaoui et al., 2007), was also determined in this study, and its concentration ranged
from 0.49 to 78 ng m$^{-3}$ (Ave. 17±17 ng m$^{-3}$). The mean concentration of
β-caryophyllinic acid in P1 is 35±21 ng m$^{-3}$, being 5 times higher than that in P2
(4.1±1.2 ng m$^{-3}$).

Undoubtedly, the combustion of biomass materials can release a large amount of

volatile organic compounds, including isoprene and terpenoids (Andreae and Merlet,
2001). As shown in Figure 5 and 6, the total biogenic SOA tracers, the sum of
detected tracers of isoprene, α-/β-pinene, and β-caryophyllene derived SOA, showed a
similar temporal variation pattern as levoglucosan with a moderate correlation
($R^2$=0.56, Figure S5a). Specifically, levoglucosan showed strong linearly correlations
with 3-hydroxyglutaric acid ($R^2$=0.72) (Figure 8c) and β-caryophyllinic acid ($R^2$=0.80)
(Figure 8d), indicating a significant contribution of biomass burning emissions to the
formation of SOA derived from mono- and sesqui- terpene oxidation. In our previous
paper (Li et al., 2018), we discussed the different diurnal variations of





isoprene-derived SOA tracers. In this study, the diurnal variations of other SOA
tracers are shown in Figure 10. All the SOA tracers presented weaker day-night
variations compared to primary organic aerosol markers, because of the competition
between the enhanced daytime formation by photoxidation and the nighttime
accumulation associated with a low PBL. Yet, there are some differences between
these SOA tracers. For example, PNA and PA presented lowest concentrations in the
afternoon (12:00-18:00 pm) due to their relatively high volatilities, which is
unfavorable for gas-to-aerosol phase partitioning. However, the later-generation
product of PNA and PA, i.e., the less volatile MBTCA, continuously increased during
the daytime.

Two classes of aromatic SOA markers, phthalic acids and

2,3-dihydroxy-4-oxopentanoic acid (DHOPA), were detected in the samples as well.
Phthalic acids are believed to be produced by the oxidation of naphthalene and other
PAHs (Kawamura et al., 2005;Kawamura and Ikushima, 1993;Kanakidou et al., 2005).
The mean concentrations of total phthalic acids in the whole sampling period ranged
from 17 to 487 ng m$^{-3}$ with a mean value of 155±94 ng m$^{-3}$. Their different temporal
variation patterns than levoglucosan suggest that biomass burning emission
contributes little to phthalic acids formation in the region. The DHOPA was
considered to be a tracer compound for toluene-derived SOA (Kleindienst et al., 2004).
DHOPA presented a similar temporal variation and moderate correlation to
levoglusoan (R$^2$=0.51, Figure S5b), indicating a certain contribution of biomass
burning. Similar to MBTCA, the volatility of DHOPA is quite low, and thus mainly



exists in the particle phase at field temperature (Ding et al., 2017). Thus, DHOPA
showed a similar diurnal variation to MBTCA, with higher concentrations during
daytime.
**3.4 Assessment of source contributions**
In order to investigate the differences in organic aerosol sources between the two
representative periods, we first classified all the measured organic compounds into
seven different sources: (a) "plant emission" represented by higher plant wax
n-alkanes, HMW fatty acids and fatty alcohols ($\geq C_{20}$); (b) "fossil fuel combustion"
mainly represented by fossil fuel derived n-alkanes, hopanes, and PAHs; (c) "biomass
burning" represented by levoglucosan and its isomers; (d) "marine/microbial source"
represented by LMW fatty acids ($<C_{20}$); (e) "soil/fungal spore/pollen" represented by
primary saccharides and sugar alcohols; (f) "plastic emission" represented by
phthalate esters; and (g) "secondary oxidation" represented by biogenic SOA tracers,
DHOPA, and phthalic acids. The concentrations of individual classes and their
contributions to OC content during P1 and P2 are summarized in Figure 11. Plant
emission-derived compounds accounted for a larger fraction of $PM_{2.5}$ OC during P1
than during P2 (mean fractions of 28.7 ± 9.3‰ in P1 vs. 16.5 ±7.2‰ in P2). The
average faction of biomass burning-derived organics in P1 was also higher in P1 than
in P2 (6.0 ± 3.9‰ vs. 4.6 ± 2.1‰), so do organics derived from soil/fungal
spore/pollen. However, organic molecules from the other 4 sources all presented a
higher contribution to OC in P2 than in P1.
While the relative abundances of organic tracer compounds in $PM_{2.5}$ offer some



insights into the relative contributions of different sources to organic aerosols, the
estimation is only qualitative because there is a potential overlapping occurrence of
many organic species from multiple sources (Simoneit et al., 2004b). In this study we
used a tracer-based source apportionment method to access the contributions of
certain primary and secondary sources to aerosol OC in the atmosphere of the rural
site. As described above, two samples collected at $21^{st}$ Jun 18:00-24:00 pm were
considered to be highly affected by the direct emission from biomass burning nearby
the sampling site. Thus, the average OC/levoglucosan ratio in the smoke of biomass
burning $\left(\left(\frac{OC}{Levo}\right)_{BB}\right)$ can be estimated by the followed equation:
$$\left(\frac{OC}{Levo}\right)_{BB} = \frac{OC_n - \frac{1}{2}(OC_{before} + OC_{after})}{Levo_n - \frac{1}{2}(Levo_{before} + Levo_{Levo})} \qquad (E1)$$

Where $OC_n$ and $Levo_n$ are the average concentrations of OC and levoglucosan in
the two samples affected by nearby sources. $OC_{before}$ and $Levo_{before}$ are the
concentrations of OC and levoglucosan in the sample before this event, whereas
$OC_{after}$ and $Levo_{after}$ are the concentrations of OC and levoglucosan in the sample after
it. The mean values in the "before" and the "after" samples were subtracted to
minimize the influence of local background contribution. The calculated $\left(\frac{OC}{Levo}\right)_{BB}$ in
this study is 18.7, which is somewhat higher than the average value of 12.3 measured
in Rondônia aerosols (Graham et al., 2002). This difference can be attributed to the
differences of biomass fuels in the two regions. For other sources, the measured
concentrations of mannitol were used to calculate the contributions of fungal spores to
OC (Bauer et al., 2008), and SOA tracers were used to estimate the SOC formed from
the oxidation of isoprene, α-/β-pinene, β-caryophyllene, and toluene (Kleindienst et



al., 2007). Also, these tracer-based approaches tend to have large uncertainties,
especially for SOC estimation (Li et al., 2013a). However, our results are still
meaningful to understand the relative abundances of organic aerosols from these
sources in different periods.
As shown in Figure 12, biomass-burning derived OC, ranging from 0.11-27.5
$\mu gC\ m^{-3}$, is the dominant source, which accounts for 1.16-74.8% (ave. 22.6%) of OC
in the aerosols of the rural region during the whole sampling period. Fungal-spore
derived OC (0.003-5.12 $\mu gC\ m^{-3}$) is a minor source, only accounting for 0.43%
(0.003-5.12%) of OC. The contribution of total SOC derived from oxidation of
isoprene, α-/β-pinene, β-caryophyllene, and toluene to OC ranged from 5.90-34.1%
with an average at 16.7%. Among the four SOC precursors, toluene-derived products
accounted for 7.78% (2.06-21.7%) of OC, being the most important SOC contributor.
The relative abundances of these sources showed clear temporal variations during the
whole sampling period (Figure 12). The contribution of biomass burning derived OC
to total OC in P1 (27.6%) was 1.7 times of that in P2 (17.1%) (Figure 13), further
indicating the strong regional impact of open-field wheat straw burning on the
molecular compositions of organic aerosols in the rural area of NCP. The contribution
of SOC from oxidation of the four VOCs increased slightly from P1 (16.3%) to P2
(21.1%). It should be noted that biomass burning can also release a large amount of
VOCs, which may produce more secondary organic aerosols during the long-range
transport. Thus, the impact of intensified biomass burning in the southern region of
NCP on organic aerosols in the Gucheng area is likely even stronger than the



estimation presented above with implications for regional climate.
**4. Summary and Conclusion**

During the entire sampling period, OC and WSOC showed strong positive

correlations with levoglucosan, and the diurnal variation of WSOC/OC was similar to
that of levoglucosan/OC, suggesting that summertime organic aerosols in the rural
area of NCP are highly affected by direct emission of BB. Higher relative abundances
and CPI values of HMW n-alkanes, fatty acids and fatty alcohols in P1 indicated an
enhancing effect of open-field biomass burning on molecular composition of organic
aerosols. PAHs, hopanes, and phthalate esters presented different temporal and diurnal
variations from levoglucosan because of the lower contribution of BB to these organic
compositions. The total biogenic SOA tracers showed a similar temporal variation and
a moderate correlation with levoglucosan, demonstrating the enhancing effect of BB
emission on BSOA formation. Later-generation SOA products, e.g., MBTCA in this
study, are unlikely affected directly by BB emission, and thus show little changes in
concentrations between the two periods. The source distribution results derived using
a tracer-based method demonstrated that the contribution of BB to organic aerosols
increased by more than 50% during the period influenced by regional open-field
biomass burning (P1) compared to the period when local emissions were more
dominant (P2). However, this contribution may even be underestimated since BB can
also release a large amount of VOCs enhancing the formation of SOA in the
atmosphere. Our results confirmed that intensified field burning of biomass fuels can
significantly influence the concentration and composition of aerosols, and thus affect





atmospheric chemistry and climate on a regional scale.

**Author Contributions**
G.H. Wang designed the experiment. G.H. Wang, T. Zhu and L.M. Zeng arranged the
sample collection. J.J. Li and G.H. Wang collected the samples. J.J. Li, G.H. Wang, J.
Li, C. Wu and W.Q. Jiang analyzed the samples. J.J. Li, and G.H. Wang performed the
data interpretation. J.J. Li, G.H. Wang and Q. Zhang wrote the paper.

**Acknowledgements**
This work was financially supported by the program from National Nature Science
Foundation of China (No. 41773117, 91543116, 41405122). The authors gratefully
acknowledge the use of fire spots data products from the Land, Atmosphere Near
real-time Capability for EOS (LANCE) system operated by the NASA/GSFC/Earth
Science Data and Information System (ESDIS) with funding provided by NASA/HQ
(https://firms.modaps.eosdis.nasa.gov/firemap/), and the NOAA Air Resources
Laboratory (ARL) for the provision of the HYSPLIT transport and dispersion model
and/or READY website (http://www.ready.noaa.gov) used in this publication.

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


Table 1 Concentrations of carbonaceous components in the time-resolved (3-h) PM$_{2.5}$ samples in
the rural site of NCP during the whole sampling period, Period 1 (P1) and Period 2 (P2).

| Component | Whole period (N=117) | | | Period 1 (N=28) | | | Period 2 (N=13) | | |
|---|---|---|---|---|---|---|---|---|---|
| | Range | Mean | SD | Range | Mean | SD | Range | Mean | SD |
| PM$_{2.5}$ (μg m$^{-3}$) | 21~395 | 159 | 89 | 133~347 | 231 | 59 | 21~62 | 43 | 14 |
| OC (μg m$^{-3}$) | 1.7~45.7 | 17.3 | 11.1 | 13.8~44.4 | 29.4 | 7.8 | 3.6~8.8 | 5.5 | 1.7 |
| EC (μg m$^{-3}$) | 0.2~22.3 | 6.5 | 4.9 | 5.3~22.3 | 12.1 | 4.0 | 0.9~2.6 | 1.5 | 0.5 |
| WSOC (μg m$^{-3}$) | 0.7~33.0 | 11.5 | 8.2 | 5.3~33.0 | 19.1 | 8.3 | 1.2~4.2 | 2.6 | 0.8 |
| WIOC (μg m$^{-3}$) | 0.3~28.1 | 6.4 | 5.1 | 4.5~28.1 | 10.3 | 4.4 | 1.2~5.5 | 3.0 | 1.3 |
| OC/EC | 1.2~7.6 | 3.0 | 0.9 | 1.9~3.2 | 2.5 | 0.4 | 2.5~5.7 | 3.8 | 1.0 |
| WSOC/OC | 0.07~0.95 | 0.63 | 0.18 | 0.30~0.85 | 0.62 | 0.16 | 0.18~0.67 | 0.48 | 0.12 |
| WIOC/OC | 0.05~0.93 | 0.37 | 0.18 | 0.15~0.70 | 0.38 | 0.16 | 0.33~0.82 | 0.52 | 0.12 |








Table 2 Average concentrations of the organic compound classes (ng m⁻³) in the time-resolved (3-h)
PM₂.₅ samples in the rural site of NCP during the whole study period, Period 1 (P1) and Period 2
(P2).

| Compounds | Whole period (N=117) | | | Period 1 (N=28) | | | Period 2 (N=13) | | |
|---|---|---|---|---|---|---|---|---|---|
| | Range | Mean | SD | Range | Mean | SD | Range | Mean | SD |
| n-Alkanes | 9.97~722.2 | 206.9 | 149.3 | 94.7~722.3 | 343.7 | 134.1 | 25.1~103.2 | 54.3 | 22.4 |
| CPI (C₁₈-C₃₆)[a] | 1.08~8.62 | 2.47 | 1.12 | 1.38~4.67 | 2.85 | 0.87 | 1.08~3.5 | 1.64 | 0.59 |
| Fatty acids | 64.6~1777 | 514.4 | 384.3 | 206.7~1528 | 900.3 | 358.3 | 81.4~234.4 | 145.3 | 47.7 |
| CPI (C₂₁:₀-C₃₀:₀)[b] | 2.26~9.15 | 4.24 | 1.14 | 3.49~6.11 | 4.21 | 0.64 | 2.26~8.57 | 3.50 | 1.64 |
| Fatty alcohols | 3.18~975.9 | 192.6 | 187.4 | 62.4~638.2 | 322.0 | 150.7 | 16.6~100.2 | 33.9 | 22.6 |
| Sugar compounds | 15.9~2228 | 432.8 | 428.9 | 151.9~1727 | 718.0 | 403.1 | 39.7~241.3 | 93.2 | 52.9 |
| galactosan (G) | 1.03~97.78 | 18.5 | 20.6 | 2.16~97.8 | 29.5 | 27.9 | 1.45~13.3 | 4.61 | 3.13 |
| mannosan (M) | 0.69~54.82 | 9.78 | 10.4 | 1.61~54.8 | 15.0 | 13.3 | 0.96~6.63 | 2.83 | 1.43 |
| levoglucosan (L) | 5.56~1447 | 240.1 | 287.8 | 29.3~1428 | 404.0 | 344.0 | 11.2~123 | 47.8 | 26.2 |
| L/M ratio | 4.03~71.8 | 22.8 | 8.85 | 13.9~71.8 | 29.7 | 12.2 | 11.3~23.1 | 18.0 | 4.28 |
| L/(G+M) ratio | 1.38~19.3 | 8.05 | 2.59 | 5.3~19.3 | 10.1 | 3.41 | 4.58~10.2 | 6.77 | 1.97 |
| PAHs | 1.11~48.5 | 12.0 | 11.0 | 4.21~37.7 | 18.6 | 11.0 | 1.25~5.01 | 2.33 | 0.98 |
| Hopanes | 0.66~10.81 | 3.46 | 2.38 | 0.86~9.97 | 4.40 | 2.48 | 1.14~2.28 | 1.81 | 0.31 |
| Phthalate esters | 17.7~219.9 | 84.9 | 41.3 | 68.8~183.1 | 111.5 | 32.7 | 31.5~100.8 | 51.1 | 18.1 |
| Phthalic acids | 17.1~487.2 | 154.5 | 93.9 | 91.3~388.6 | 211.0 | 87.1 | 17.1~81 | 46.3 | 17.1 |
| Isoprene SOA tracers | 11.1~404.1 | 111.9 | 85.8 | 48.3~404.1 | 208.5 | 104.9 | 34.8~127.5 | 57.0 | 29.4 |
| Monoterpene SOA tracers | 11.1~166.2 | 66.1 | 31.2 | 37.3~166.2 | 85.3 | 34.9 | 26.7~64.5 | 44.6 | 12.6 |
| β-Caryophyllinic acid[c] | 0.49~77.7 | 17.4 | 17.1 | 4.6~77.8 | 34.7 | 20.8 | 2.44~6.28 | 4.08 | 1.21 |
| DHOPA[d] | 1.59~35.3 | 9.36 | 7.15 | 4.06~35.3 | 15.6 | 9.80 | 2.7~6.99 | 4.16 | 1.42 |
| Total measured organics | 176.9~6249 | 1806 | 1308 | 843.3~5499 | 2973 | 1219 | 334.2~913.7 | 537.9 | 151.1 |
| Total organics_C/OC[e] (%) | 3.19~16.0 | 6.99 | 1.97 | 3.43~8.86 | 6.43 | 1.36 | 3.77~8.61 | 6.41 | 1.27 |

[a] CPI (C₁₈-C₃₆): carbon preference index for n-alkanes, (C₁₉+C₂₁+C₂₃+C₂₅+C₂₇+C₂₉+C₃₁+C₃₃+C₃₅)/
(C₁₈+C₂₀+C₂₂+C₂₄+C₂₆+C₂₈+C₃₀+C₃₂+C₃₄).
[b] CPI (C₂₁:₀-C₃₀:₀): carbon preference index for fatty acids, (C₂₂:₀+C₂₄:₀+C₂₆:₀+C₂₈:₀+C₃₀:₀)/ (C₂₁:₀+C₂₃:₀+C₂₅:₀+C₂₇:₀+C₂₉:₀).
[c] β-Caryophyllinic acid: a tracer of β-caryophyllene-derived SOA.
[d] DHOPA: 2,3-dihydroxy-4-oxopentanoic acid, a tracer of toluene-derived SOA.
[e] All the quantified organic compounds were converted to their carbon contents to calculate the OC ratios.





# Figure Captions

Figure 1. Backward trajectories of air masses (a,c) (provided by NOAA HYSPLIT modeling system,
http://ready.arl.noaa.gov/HYSPLIT.php), and fire spots (b,d) (provided by Fire Information
for Resource Management System, FIRMS, https://firms.modaps.eosdis.nasa.gov/firemap/),
during Period 1 (P1) (Jun 13th 21:00-16th 15:00, 2013) and Period 2 (P2) (Jun 22nd 12:00-24th
06:00, 2013). Sampling site represented as purple star.

Figure 2. Temporal variations of $PM_{2.5}$, OC, EC, and WSOC during the whole sampling period.
Shadows denote the two representative periods.

Figure 3. Linear correlations of OC with WSOC (a), levoglucosan with OC and WSOC(b).
Figure 4. Diurnal variation of OC/EC (a), WSOC/OC and levoglucosan/OC (b).
Figure 5. Temporal variations of ten organic compound classes detected in the summertime $PM_{2.5}$
samples at the rural site of NCP.

Figure 6. Temporal variations of organic tracers for biomass burning (a), and secondary products
derived from $\alpha$-/$\beta$-pinene (b-d), $\beta$-caryophyllene (e), and toluene (f).

Figure 7. Molecular distributions of $n$-alkanes (a and d), fatty acids (b and e), and fatty alcohols (c and
f) in the $PM_{2.5}$ of the rural area.

Figure 8. Linear correlations of fatty acids (a), fatty alcohols (b), 3-hydroxyglutaric acid (c), and
$\beta$-caryophyllinic acid (d) with levoglucosan.

Figure 9. Diurnal variation of the detected organic compound classes.
Figure 10. Diurnal variation of the SOA tracers derived from oxidation of $\alpha$-/$\beta$-pinene (a-d),
$\beta$-caryophyllene (e), and toluene (f).

Figure 11. A comparison of the average contributions of different sources-derived organics (converted
to carbon content) to OC during P1 and P2.

Figure 12. Contributions (above) of primary organic carbon from biomass burning ($OC_{bb}$) and fungal
spores ($OC_{fp}$), and secondary organic carbon from isoprene ($SOC_i$), $\alpha$-/$\beta$-pinene ($SOC_p$),
$\beta$-caryophyllene ($SOC_p$), and toluene ($SOC_t$) to OC in the time-resolved (3 h) rural aerosols,
and their relative abundances (down). All the contributions were estimated by tracer-based
method.

Figure 13. Average contributions of direct emissions from biomass burning (BB) and fungal spores
($OC_{fp}$), secondary oxidation from isoprene ($SOC_i$), $\alpha$-/$\beta$-pinene ($SOC_p$), $\beta$-caryophyllene
($SOC_p$), and toluene ($SOC_t$) to OC in P1 and P2. All the contributions were estimated by
tracer-based method.








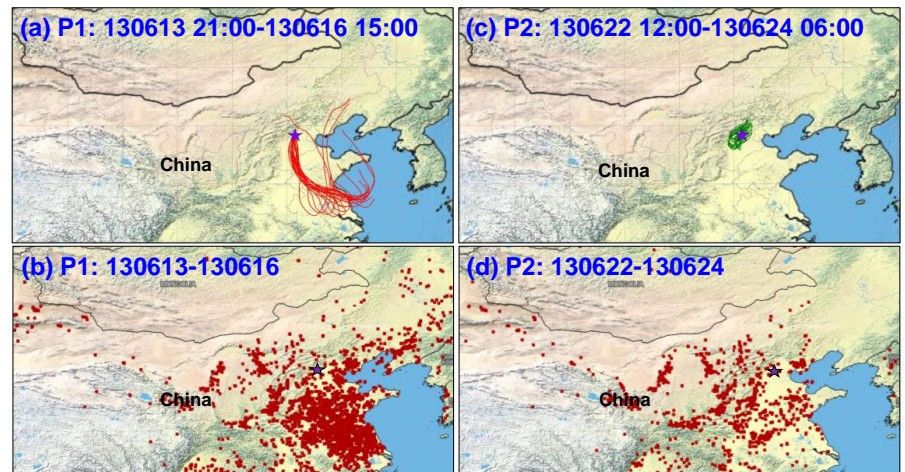


Figure 1. Backward trajectories of air masses (a,c) (provided by NOAA HYSPLIT modeling system, http://ready.arl.noaa.gov/HYSPLIT.php), and fire spots (b,d) (provided by Fire Information for Resource Management System, FIRMS, https://firms.modaps.eosdis.nasa.gov/firemap/), during Period 1 (P1) (Jun 13th 21:00-16th 15:00, 2013) and Period 2 (P2) (Jun 22nd 12:00-24th 06:00, 2013). Sampling site represented as purple star.






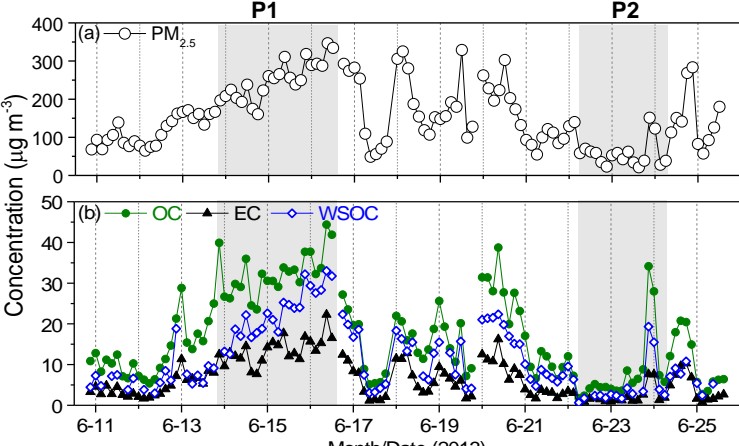


Figure 2. Temporal variations of PM$_{2.5}$, OC, EC, and WSOC during the whole sampling period. Shadows denote the two representative periods.







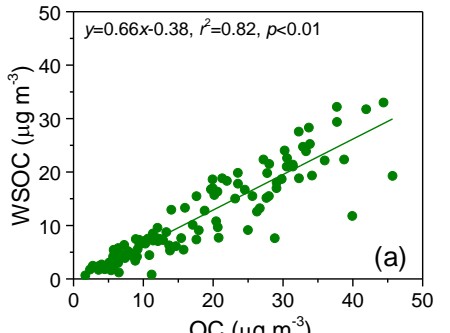
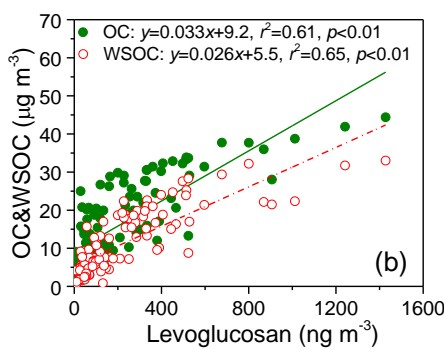


Figure 3. Linear correlations of OC with WSOC (a), levoglucosan with OC and WSOC(b).



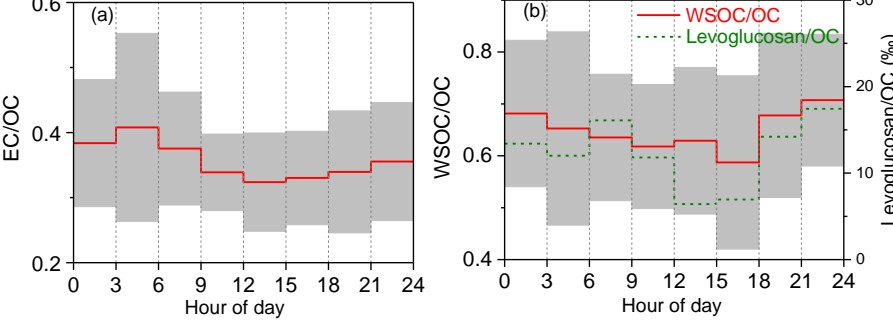


Figure 4. Diurnal variation of OC/EC (a), WSOC/OC and levoglucosan/OC (b).



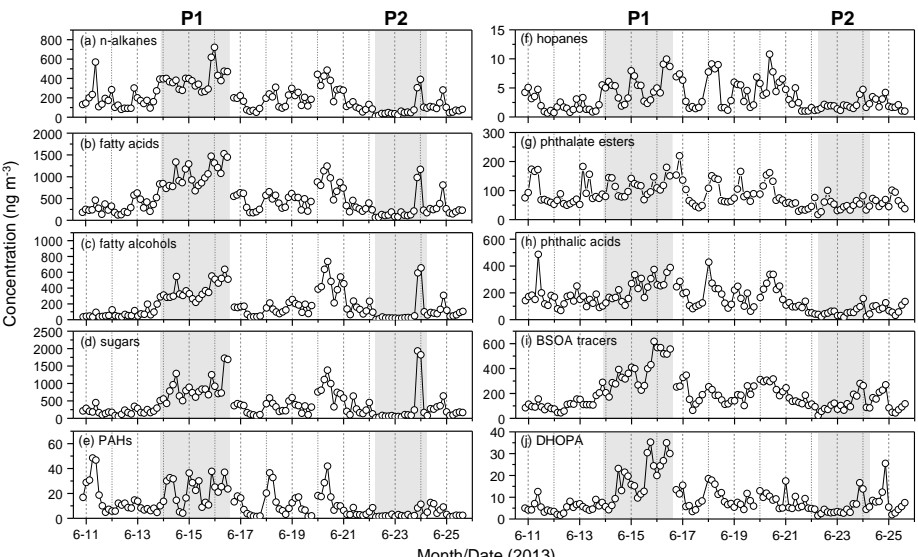


Figure 5. Temporal variations of ten organic compound classes detected in the summertime $PM_{2.5}$
samples at the rural site of NCP.






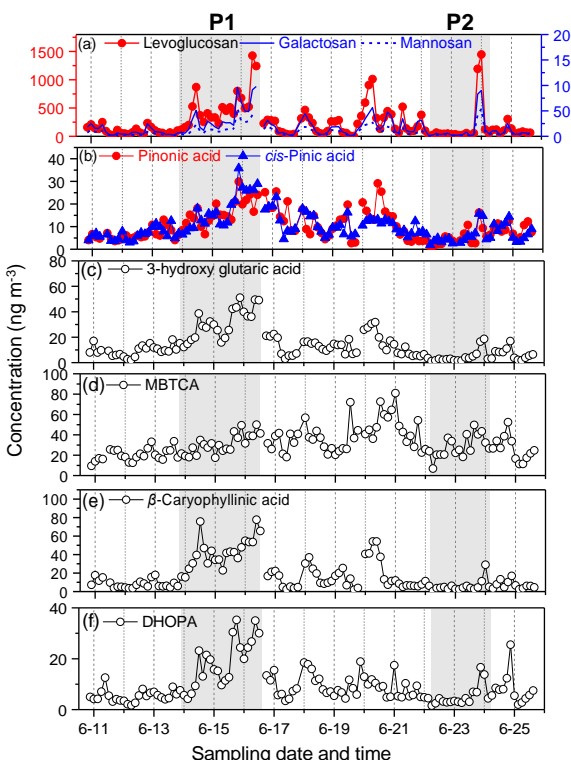


Figure 6. Temporal variations of organic tracers for biomass burning (a), and secondary products
derived from $\alpha$-/$\beta$-pinene (b-d), $\beta$-caryophyllene (e), and toluene (f).



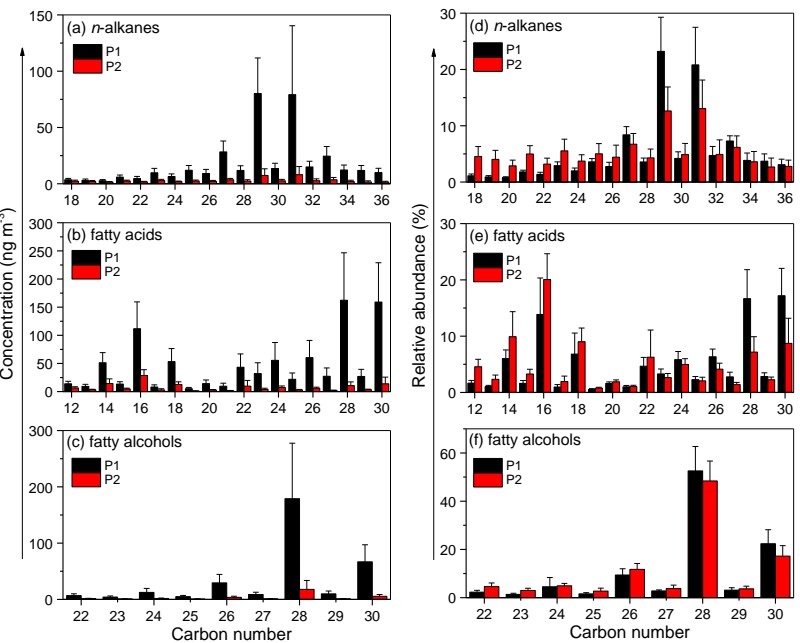


Figure 7. Molecular distributions of *n*-alkanes (a and d), fatty acids (b and e), and fatty alcohols (c and
f) in the PM$_{2.5}$ of the rural area.

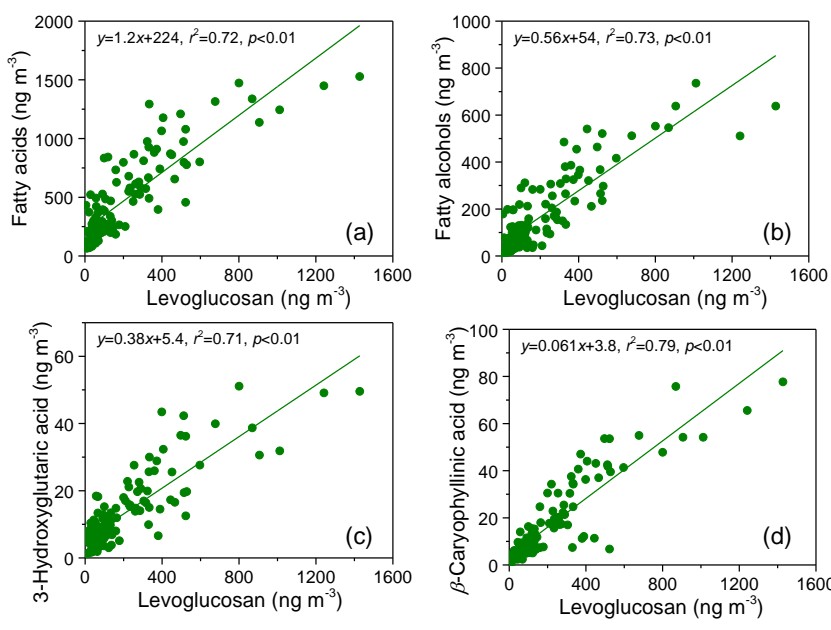


Figure 8. Linear correlations of fatty acids (a), fatty alcohols (b), 3-hydroxyglutaric acid (c), and
β-caryophyllinic acid (d) with levoglucosan.




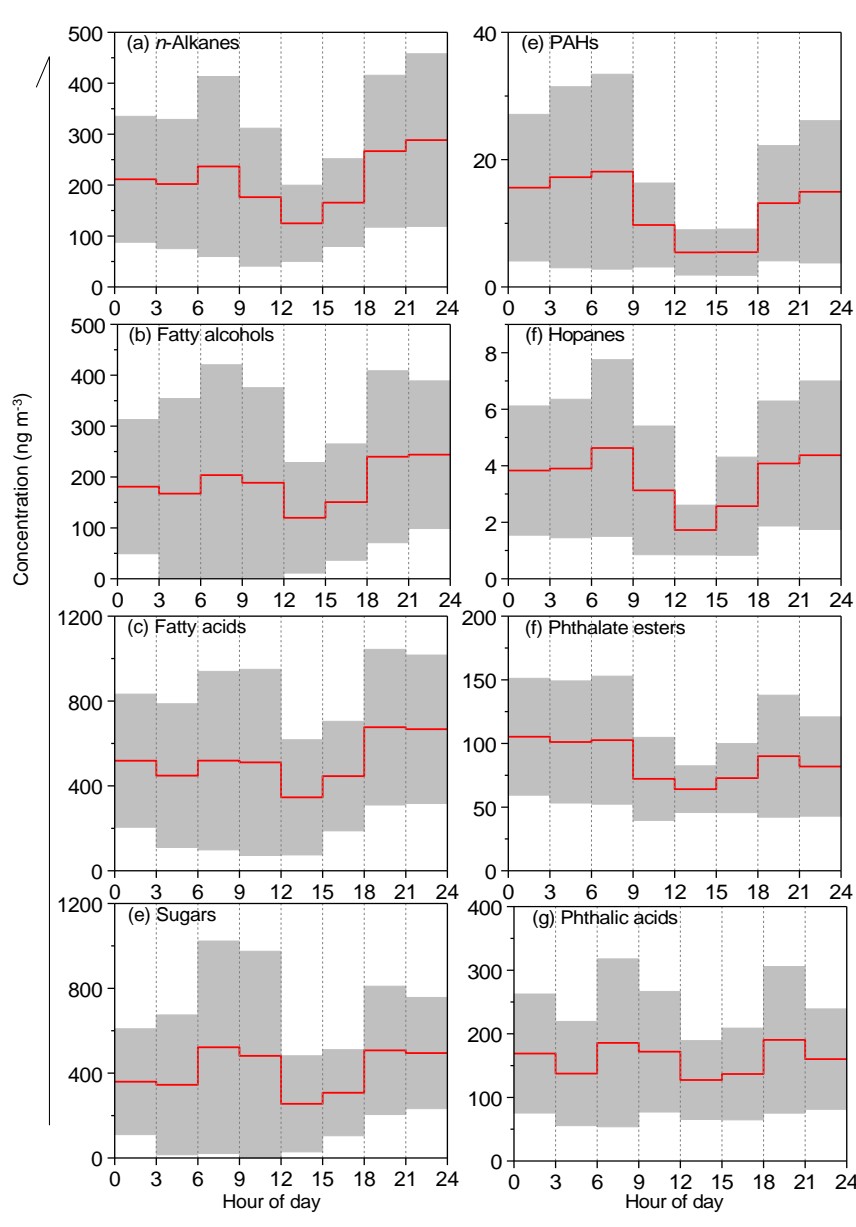


Figure 9. Diurnal variation of the detected organic compound classes.









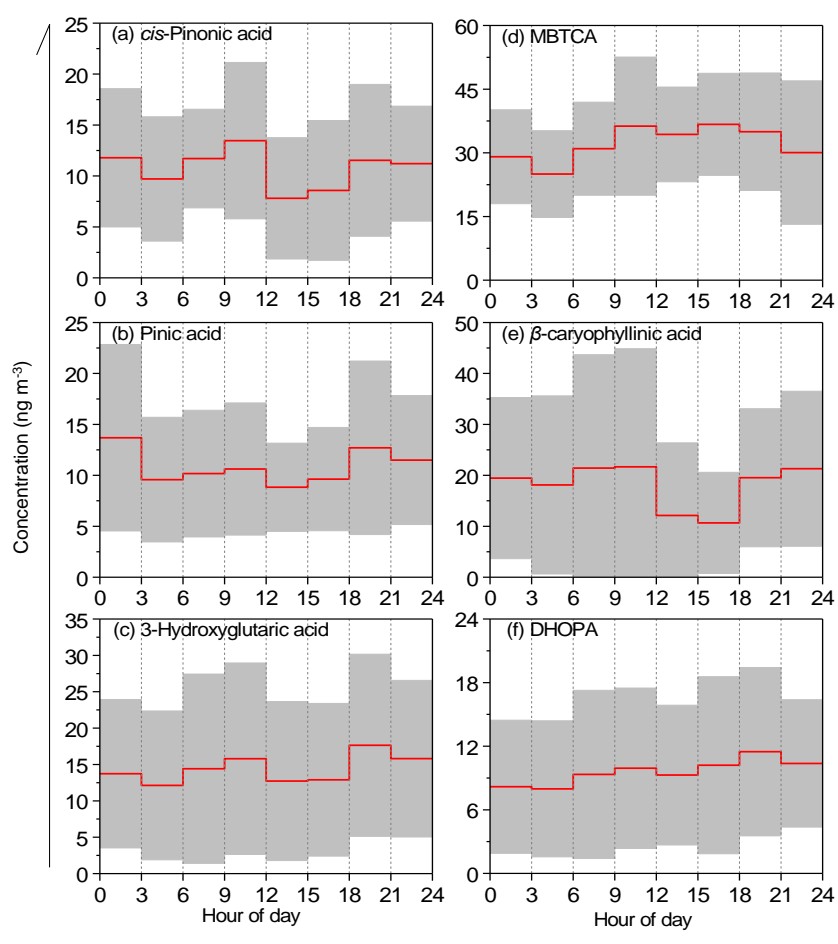


Figure 10. Diurnal variation of the SOA tracers derived from oxidation of $\alpha$-/$\beta$-pinene (a-d),
$\beta$-caryophyllene (e), and toluene (f).




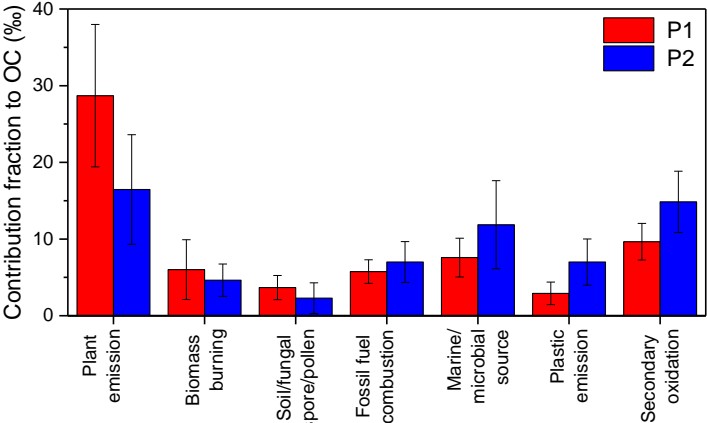

Figure 11 A comparison of the average contributions of different sources-derived organics (converted
to carbon content) to OC during P1 and P2.

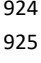

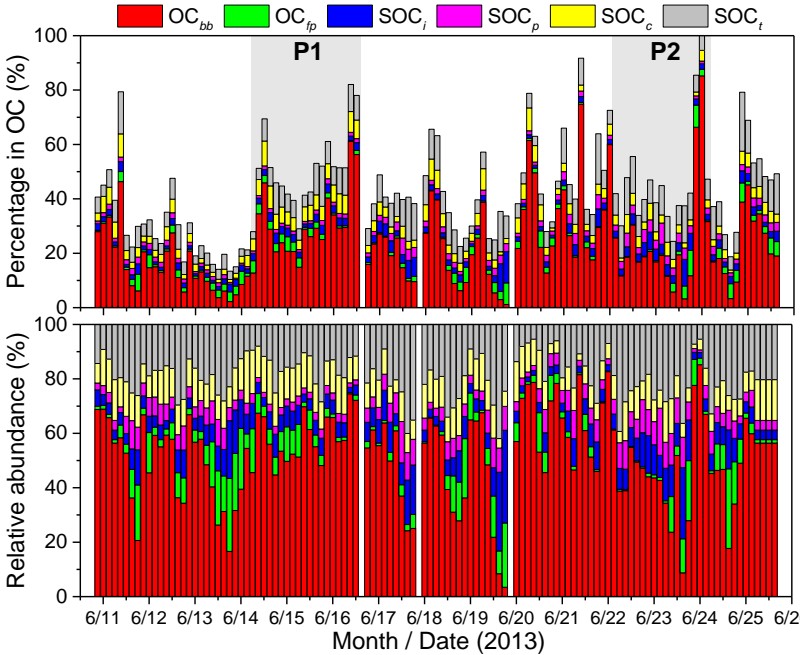

Figure 12. Contributions (above) of primary organic carbon from biomass burning (OC$_{bb}$) and fungal
spores (OC$_{fp}$), and secondary organic carbon from isoprene (SOC$_i$), α-/β-pinene (SOC$_P$),
β-caryophyllene (SOC$_P$), and toluene (SOC$_t$) to OC in the time-resolved (3 h) rural aerosols, and their
relative abundances (down). All the contributions were estimated by tracer-based method.




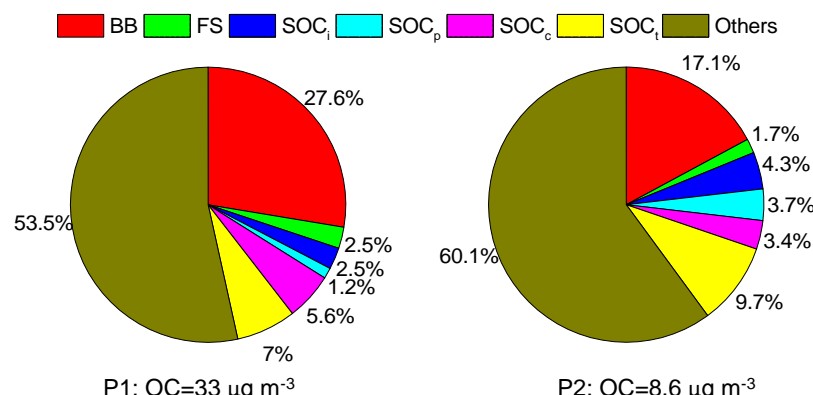


Figure 13. Average contributions of direct emissions from biomass burning (BB) and fungal spores
(OC$_{fp}$), secondary oxidation from isoprene (SOC$_i$), $\alpha$-/$\beta$-pinene (SOC$_p$), $\beta$-caryophyllene (SOC$_p$), and
toluene (SOC$_t$) to OC in P1 and P2. All the contributions were estimated by tracer-based method.
