# Peer review of "Molecular characteristics and diurnal variations of organic aerosols"

_Atmospheric Chemistry and Physics, 2019_

## Referee Comment (RC1) · Anonymous Referee #1 · 1 Jun 2019

The manuscript by Li et al. had a comprehensive characterization of molecular composition of organic aerosol at a rural site in North China Plain, and investigated the impacts of biomass burning, and the sources of organic aerosol. One advantage of this study is the high time resolution of filter sampling (3 h) compared with previous studies. The authors chose two different periods to discuss the differences between local and regional sources. Overall, this study presents a huge dataset containing rich chemical information, and it is worth for publication in ACP.

My major concern is the selection of period 2 (P2). The time series of species showed

that P2 was characterized by a clean period (winds dominantly from the north, see supplementary) followed by a strong local event (very likely). This indicated a mixed event rather than a pure local. In fact, the chosen of P2 can affect the conclusion substantially. I suggest the authors reevaluating the event of P2.

Biomass burning typically showed much higher OC/EC ratios than those reported in this study, could the authors give more explanation?

---

## Referee Comment (RC2) · Anonymous Referee #3 · 26 Jun 2019

This manuscript presents measurement results from a field study conducted in the North-China Plain, which is notorious for high aerosol pollution. While suffering slightly from a relatively short measurement period, this study presents an impressive suite of organic aerosol component concentrations at a rather high time resolution. Two specific periods were singled out, including one episode with high biomass smoke impact. The results presented in this paper are helpful for a better understanding of the sources and characteristics of organic aerosols in this highly polluted part of China. Prior to publication of the manuscript in ACP, the authors should address the comments and

suggestions below.

Specific comments:

1. Line 136: The total sampling period is rather short, especially when dividing it into 2 special periods, requiring caution in the discussion of the measurement results. The authors may want to add a statement regarding how representative the data are.

2. Lines 161-163: The authors corrected the data for the field blanks, although the blank values were relatively low, in contrast to the recoveries which introduced larger errors for certain species. Why were the recoveries not taken into account as well?

3. Lines 220-223: Do the authors have a possible explanation for the rather low OC/EC ratios measured during the biomass burning period? Previous studies, especially those investigating burns which were dominated by smoldering combustion, were characterized by emissions with significantly higher OC/EC ratios. Is it possible that the wheat straw combustion during the study period occurred to some extent in the flaming phase?

4. Lines 241-245: Indeed, the regional biomass burning activities contributed to the elevated WSOC/OC fractions, but it may also be worthwhile mentioning here (as the authors do later on in the paper) that SOA was likely produced in the biomass burning plumes (especially considering the transport distance/time to the sampling site), and thus contributed to the higher degree of oxygenation of the organic aerosol as well.

5. Lines 287-288: It would be helpful for the readers who are not familiar with these diagnostic ratios to at least briefly explain how the high L/M ratios indicate straw burning.

6. Lines 288-289: How are the anhydrosugar emission ratios of lignites relevant to this study? Wouldn't it be more useful to mention results from some of the previous studies which specifically investigated anhydrosugar emissions from burning of straw or similar types of biomass?

7. Lines 290-294: It would be helpful if the authors showed more quantitative results,

e.g., state what is considered "higher". And how specifically do these results confirm the contribution of wheat straw burning?

8. Lines 503-505: Why do the authors mention measurements from this area, as there seems to be no relation to this study region? Why not show data from other areas in Asia?

Technical corrections:

1. Lines 61, 199, 491: A better expression for "access" might be "estimate".

2. Lines 106, 107, 295, 296: Use correct singular vs. plural forms of words throughout the manuscript, such as "straw" instead of "straws", "amounts" instead of "amount", "composition" instead of "compositions" and "concentration" instead of "concentrations", respectively.

3. Lines 137: Please specify if a size-selective inlet was used on the Hi-vol or if total suspended particles (TSP) were collected.

4. Lines 190 and 196: The definite article "the" before "North China" and "wheat" is not needed.

5. Lines 239-240: Shouldn't the favorable conditions for photo-chemical oxidation result in higher WSOC/OC values?

6. Line 265: Change "ranged" to "ranging".

7. Lines 333-337: This sentence is not easy to understand and therefore needs to be reworded.

8. Lines 350-352: Another study by Yang et al (STOTEN, 2012) specifically addresses this phenomenon.

9. Lines 385-386: These correlations are not week but there is no correlation.

10. Lines 392-395: Revise the sentence as follows: "..., consistent with the favorable

dispersion conditions caused by high temperatures and planetary boundary layer (PBL) height.

11. Line 451: Change "aerosol" to "particle".

12. Lines 503-505: Are these % values?

---

## Author Comment (AC1) · 18 Jul 2019

Journal: ACP
Title: Molecular characteristics and diurnal variations of organic aerosols at a rural
site in the North China Plain with implications for the influence of regional biomass
burning
Author(s): Jianjun Li et al.
MS No.: acp-2019-75

Dear Editor,

After reading the comments from the two referees, we have carefully revised our

manuscript. Our response to the comments and related revisions are attached with this

letter.

Anything about our paper, please feel free to contact me at ghwang@geo.ecnu.edu.cn.

Best regards,

Sincerely yours

Gehui Wang

Jul. 18, 2019

**Anonymous Referee #1:**

The manuscript by Li et al. had a comprehensive characterization of molecular composition of organic aerosol at a rural site in North China Plain, and investigated the impacts of biomass burning, and the sources of organic aerosol. One advantage of this study is the high time resolution of filter sampling (3 h) compared with previous studies. The authors chose two different periods to discuss the differences between local and regional sources. Overall, this study presents a huge dataset containing rich chemical information, and it is worth for publication in ACP.

My major concern is the selection of period 2 (P2). The time series of species showed that P2 was characterized by a clean period (winds dominantly from the north, see supplementary) followed by a strong local event (very likely). This indicated a mixed event rather than a pure local. In fact, the chosen of P2 can affect the conclusion substantially. I suggest the authors reevaluating the event of P2.

Biomass burning typically showed much higher OC/EC ratios than those reported in this study, could the authors give more explanation?

**Response:** We thank the referee's comments.

For the selection of the period 2 (P2). We agree that it's not easy to select a period that is representative of a pure local emission in field study. However, in this study, the 72 h backward trajectories clearly show that air masses came predominantly from the surrounding areas of the sampling site during P2 (Figure 1). Although the winds in sometimes from the north, the wind speed in most time of P2 are <2 m s$^{-1}$ (Figure S2). In addition, all the chemical compositions present very low concentrations during P2 except 2 samples affected by a near-site biomass burning emission. These results confirm that local fresh emissions dominated the chemical compositions of aerosols during P2.

For the OC/EC ratios, as suggested by the referee, we compared the OC/EC ratios in this study with those reported by some chamber or filed studies. We concluded that the relatively low OC/EC ratio in our study is related to more complex source contributions and different combustion conditions of open biomass burning. More explanation can be found in Line 225-237.

**Anonymous Referee #3:**

This manuscript presents measurement results from a field study conducted in the North-China Plain, which is notorious for high aerosol pollution. While suffering slightly from a relatively short measurement period, this study presents an impressive suite of organic aerosol component concentrations at a rather high time resolution. Two specific periods were singled out, including one episode with high biomass smoke impact. The results presented in this paper are helpful for a better understanding of the sources and characteristics of organic aerosols in this highly polluted part of China. Prior to publication of the manuscript in ACP, the authors should address the comments and suggestions below.

**Response:** We thank the referee's comments, which are very helpful for us to improve our work. Detailed revision and response to the comments are list below.

**Specific comments:**

1. Line 136: The total sampling period is rather short, especially when dividing it into 2 special periods, requiring caution in the discussion of the measurement results. The authors may want to add a statement regarding how representative the data are.

**Response:** Suggestion taken. We revised related description, please see Line 134-140.

2. Lines 161-163: The authors corrected the data for the field blanks, although the blank values were relatively low, in contrast to the recoveries which introduced larger errors for certain species. Why were the recoveries not taken into account as well?

**Response:** In our study, averaged recoveries of the target compounds were better than 70%. Recovery experiment is a method for QA and QC. However, compounds used in a recovery experiment are usually pure agent while those in real samples are a mixture with other organic and inorganic components, which means that the recovery experiment could not entirely reflect the conditions of target compounds in the atmosphere. Thus, many documents report the data without a correction by recovery. For example, US ASTM method D 6209-98 for atmospheric PAH, the section 16.4.2

at page 12 notes that "Typically, measured PAH analyte concentrations are not corrected for surrogate recovery". Therefore, in this paper the data reported were not corrected by the recoveries.

3. Lines 220-223: Do the authors have a possible explanation for the rather low OC/EC ratios measured during the biomass burning period? Previous studies, especially those investigating burns which were dominated by smoldering combustion, were characterized by emissions with significantly higher OC/EC ratios. Is it possible that the wheat straw combustion during the study period occurred to some extent in the flaming phase?
**Response:** By comparing the OC/EC ratios in this study with those reported in some chamber or filed studies, we do agree with the referee's comment that wheat straw combustion during the study period occurred to some extent in the flaming phase. On the other hand, we think the contribution of fossil fuel combustion in NCP is also a reason for lower OC/EC ratio in this study. Thus, we added more explanation at Line 225-237 in the manuscript.

4. Lines 241-245: Indeed, the regional biomass burning activities contributed to the elevated WSOC/OC fractions, but it may also be worthwhile mentioning here (as the authors do later on in the paper) that SOA was likely produced in the biomass burning plumes (especially considering the transport distance/time to the sampling site), and thus contributed to the higher degree of oxygenation of the organic aerosol as well.
**Response:** We do agree with the referee's comment, and revised the related description as "many SOA could be produced in the biomass burning plumes during the long-range transport (detailed discussions are given in Section 3.3), these results indicate that particulate WSOC in the region is mostly contributed from biomass burning activities including direct emission and secondary oxidation" (Line 256-260).

5. Lines 287-288: It would be helpful for the readers who are not familiar with these diagnostic ratios to at least briefly explain how the high L/M ratios indicate straw burning.
**Response:** Suggestion taken. We gave an explanation at Line 302-307.

6. Lines 288-289: How are the anhydrosugar emission ratios of lignites relevant to this study? Wouldn't it be more useful to mention results from some of the previous studies which specifically investigated anhydrosugar emissions from burning of straw or similar types of biomass?

**Response:** We thank the referee's comment. We clarified the statement as " Fabbri et al. (2009) compared the concentrations of the three anhydrosaccharides in the smokes from different fuel types, and proposed that levoglucan/(galactosan+mannosan) (L/G+M) and levoglucan/mannosan (L/M) values range in 0.2-18 and 0.23-33 for various source tests for biomass burning as compared to the average of 54 and 54 for lignites" (Line 303-307).

7. Lines 290-294: It would be helpful if the authors showed more quantitative results, e.g., state what is considered "higher". And how specifically do these results confirm the contribution of wheat straw burning?

**Response:** Suggestion taken. We added quantitative results of L/G+M and L/M ratios in the statement, and reevaluated results as "average ratios of L/G+M and L/M during P1 ($10.1\pm3.41$ and $6.77\pm1.97$, respectively) and P2 ($29.7\pm12.2$ and $18.0\pm4.28$) suggest that biomass burning is always the dominated contributor for these compounds in NCP" (Line 307-310).

8. Lines 503-505: Why do the authors mention measurements from this area, as there seems to be no relation to this study region? Why not show data from other areas in Asia?

**Response:** Suggestion taken. We revised the related discussion, and cited a new data measured in the $PM_{2.5}$ samples emitted from burning of three main kinds of cereal straws (wheat, maize, and rice) in China. Please see Line 517-519.

**Technical corrections:**

1. Lines 61, 199, 491: A better expression for "access" might be "estimate".

**Response:** Suggestion taken. Please see Line 60, 201, 504.

2. Lines 106, 107, 295, 296: Use correct singular vs. plural forms of words throughout the manuscript, such as "straw" instead of "straws", "amounts" instead of "amount", "composition" instead of "compositions" and "concentration" instead of

"concentrations", respectively.

**Response:** Suggestion taken. We have carefully checked the grammar mistakes thorough the whole manuscript, such as Line 106, 107, 311, 312.

3. Lines 137: Please specify if a size-selective inlet was used on the Hi-vol or if total suspended particles (TSP) were collected.

**Response:** Suggestion taken. Please see Line 138-139.

4. Lines 190 and 196: The definite article "the" before "North China" and "wheat" is not needed.

**Response:** Suggestion taken. Please see Line 192 and 198.

5. Lines 239-240: Shouldn't the favorable conditions for photo-chemical oxidation result in higher WSOC/OC values?

**Response:** Indeed, the favorable conditions during afternoon for photo-chemical oxidation should result in higher WSOC/OC values. However, in this study, we found that WSOC/OC value presents lower value in the afternoon. On the other hand, we found diurnal variation pattern of WSOC/OC is similar to that of levoglucosan/OC. Thus, we concluded that WSOC/OC value in the study area is mainly affected by the transportation of biomass burning. Related discussion please see Line 252-260.

6. Line 265: Change "ranged" to "ranging".

**Response:** Suggestion taken. Please see Line 280.

7. Lines 333-337: This sentence is not easy to understand and therefore needs to be reworded.

**Response:** We revised the sentence, please see Line 348-352.

8. Lines 350-352: Another study by Yang et al (STOTEN, 2012) specifically addresses this phenomenon.

**Response:** We thank the referee's comment, and cited the reference at Line 368.

9. Lines 385-386: These correlations are not week but there is no correlation.

**Response:** We revised the sentence, please see Line 401-402.

10. Lines 392-395: Revise the sentence as follows: "..., consistent with the favorable dispersion conditions caused by high temperatures and planetary boundary layer (PBL) height.

**Response:** Suggestion taken. Please see Line 410-411.

11. Line 451: Change "aerosol" to "particle".

**Response:** Suggestion taken. Please see Line 465.

12. Lines 503-505: Are these % values?

**Response:** They are absolute values, not % values.

---

## Author Response (AR2)

Journal: ACP
Title: Molecular characteristics and diurnal variations of organic aerosols at a rural site in the North China Plain with implications for the influence of regional biomass burning
Author(s): Jianjun Li et al.
MS No.: acp-2019-75

Dear Editor,

We thank you very much for the comments. In this manuscript, we checked the text for grammar mistakes carefully. Please see the details in the following marked-up manuscript.

Anything about our paper, please feel free to contact me at ghwang@geo.ecnu.edu.cn.

Best regards,

Sincerely yours

Gehui Wang

Jul. 28, 2019

[revised manuscript text omitted]